# Accurate and Lightweight Learning for Specific Domain Image-Text Retrieval

## ABSTRACT

Recent advances in vision-language pre-trained models like CLIP have greatly enhanced general domain image-text retrieval performance. This success has led scholars to develop methods for applying CLIP to Specific Domain Image-Text Retrieval (SDITR) tasks such as Remote Sensing Image-Text Retrieval (RSITR) and Text-Image Person Re-identification (TIReID). However, these methods for SDITR often neglect two critical aspects: the enhancement of modal-level distribution consistency within the retrieval space and the reduction of CLIP's computational cost during inference, resulting in suboptimal retrieval spaces and unnecessarily high inference computational loads. To address these issues, this paper presents a novel framework, **A**ccurate and lightweight learning for specific domain **I**mage-text **R**etrieval (AIR), based on the CLIP architecture. AIR incorporates a Modal-Level distribution Consistency Enhancement regularization (MLCE) loss and a Self-Pruning Distillation Strategy (SPDS) to improve retrieval precision and computational efficiency. The MLCE loss harmonizes the sample distance distributions within image and text modalities, fostering a retrieval space closer to the ideal state. Meanwhile, SPDS employs a strategic knowledge distillation process to transfer deep multimodal insights from CLIP to a shallower level, maintaining only the essential layers for inference, thus achieving model light-weighting. Comprehensive experiments across various datasets in RSITR and TIReID demonstrate the effectiveness of both MLCE loss and SPDS. The study also explores the limits of SPDS's performance and compares it with conventional teacher-student distillation methods. The findings reveal that MLCE loss secures optimal retrieval on several datasets, while SPDS achieves a favorable balance between accuracy and computational demand during testing.

## CCS CONCEPTS

• **Information systems** → **Multimedia and multimodal retrieval**; • **Computing methodologies** → **Neural networks**.

## KEYWORDS

Cross-modal image-text retrieval, Lightweight, Vision-language pre-trained models, Remote sensing, Text-image person re-identification

**ACM Reference Format:**
Anonymous Authors. 2024. Accurate and Lightweight Learning for Specific Domain Image-Text Retrieval. In *Proceedings of the 32nd ACM International Conference on Multimedia (MM'24), October 28-November 1, 2024, Melbourne, Australia.* ACM, New York, NY, USA, 10 pages. https://doi.org/10.1145/nnnnnnn.nnnnnnn

## 1 INTRODUCTION

Cross-modal image-text retrieval (CMITR) enables semantic-based matching between text and images and has garnered significant attention from scholars [11, 25, 56, 58, 69]. CMITR can be dichotomized based on its application scope: General Domain Image-Text Retrieval (GDITR) addresses natural or everyday scenes [5, 21, 22], while Specific Domain Image-Text Retrieval (SDITR) focuses on specialized fields, such as Remote Sensing Image-Text Retrieval (RSITR) [42, 68, 72] and Text-Image person Re-identification (TIReID) [19, 52, 66]. In the realm of GDITR, there is a growing consensus on the efficacy of employing Visual-Language large-scale Pre-training models (VLPs) [2, 10, 32, 47, 48]. A typical strategy involves fine-tuning VLPs on targeted GDITR datasets [13]. This methodology has been applied with considerable success in GDITR, leading to significant enhancements in performance metrics [17, 24, 27, 30, 46]. Notably, CLIP [46], as an exemplary VLP, underscores this progress. Trained on a diverse dataset of 400 million web-based image-text pairs, CLIP has established a new standard for image-text alignment and spurred the creation of novel approaches that have propelled GDITR accuracy to unprecedented levels.

The remarkable achievements of CLIP in GDITR have prompted researchers to extend its utility to SDITR. One straightforward method of adaptation is full fine-tuning on SDITR datasets [19]. Furthermore, several researchers have delved into alternative approaches, such as prompt learning [23] and the integration of adaptors [71], which have significantly advanced CLIP's efficacy in SDITR. Despite these improvements, current methods have not adequately addressed two critical concerns.

Firstly, existing methods primarily employ contrastive loss to focus on instance-level cross-modal alignment, yet they overlook the importance of maintaining modal-level distribution consistency within the joint representation space in SDITR. Modal-level distribution consistency refers to the uniformity between the distance distributions of samples within different modalities. The better the alignment between image and text modalities, the higher the modal-level distribution consistency observed. Figure 1 (a) illustrates an ideal state of maximal modal-level distribution consistency. We quantify modal-level distribution consistency using the Kullback-Leibler (KL) divergence between image and text intramodal self-similarity matrices. Comparative evaluations on GDITR and SDITR tasks (Figure 1(e)) reveal a positive correlation between modal-level distribution consistency and retrieval performance, with SDITR showing notably lower precision and weaker modal-level distribution consistency. Meanwhile, there is an inherent lack of modal-level distribution consistency in the SDITR task data, as evidenced in Figure 1 (b)-(d), indicating the necessity for a specific focus on enhancing modal-level distribution consistency in

SDITR tasks. However, current adaptations of CLIP for SDITR do not adequately resolve this inconsistency. Consequently, this work advocates that the enhancement of modal-level distribution consistency can facilitate the formation of an optimal retrieval space in SDITR, thereby achieving more accurate retrieval results.

Secondly, the demands of real-time processing in SDITR are not adequately addressed, as these methods neglect the imperative of lightweight inference for efficient data stream processing. Tasks in SDITR often necessitate high real-time performance. For example, RSITR typically involves processing an extensive gallery, necessitating both lightweight inference and real-time processing capabilities [73]. Furthermore, the task of semantic localization through RSITR, which must handle a substantial volume of large-scale remote sensing images, also requires robust real-time performance [73]. In TIReID, particularly within security and surveillance applications, the need for swift inference is paramount [19]. Existing methods [71] have focused primarily on diminishing training overheads without considering the computational and temporal efficiencies during inference. Consequently, to facilitate the application of CLIP in SDITR, optimizing for lightweight retrieval during testing is essential to streamline the data stream.

To tackle the issues previously outlined, we introduce a novel framework termed **A**ccurate and lightweight learning for Specific Domain **I**mage-text **R**etrieval (AIR), grounded in the CLIP architecture. AIR integrates a Modal-Level distribution Consistency Enhancement regularization (MLCE) loss and a Self-Pruning Distillation Strategy (SPDS), designed to refine retrieval performance while maintaining computational efficiency.

The MLCE loss is engineered to bolster the uniformity of intra-modal distributions for both images and texts, thereby augmenting modal-level distribution consistency and guiding the optimization of the shared representation toward an ideal retrieval space. This approach is instrumental in cultivating refined joint image-text representations tailored for SDITR, while simultaneously reinforcing modal interactions within CLIP. In practice, the MLCE loss reduces the KL divergence between the self-similarity matrices of the image modality and the text modality within a mini-batch, fostering consistency in the distribution of sample distances within different modalities. This optimization is directed towards enhancing the joint representation to align with the ideal scenario depicted in Figure 1 (a).

SPDS, drawing on the principles of knowledge distillation, facilitates self-pruning by harnessing intra-CLIP knowledge transfer. The streamlined model preserves robust retrieval capabilities while curtailing the parameter count and computational demands. SPDS uses the image-text similarity matrix from CLIP's final layer as a pedagogical guide for the matrices derived from the initial K layers. This training approach effectively redistributes the deeper layers' learned knowledge to earlier stages. During inference, the retention of only the first K layers in the image (text) encoder's transformer substantially reduces the model's complexity.

Our extensive empirical analysis across two SDITR tasks—RSITR and TIReID—demonstrates that AIR achieves both precise and efficient retrieval outcomes. The utilization of MLCE loss in CLIP results in an mR retrieval metric that outperforms existing approaches, thereby setting a new state-of-the-art (SOTA) benchmark and corroborating the efficacy of the MLCE loss. Moreover, we

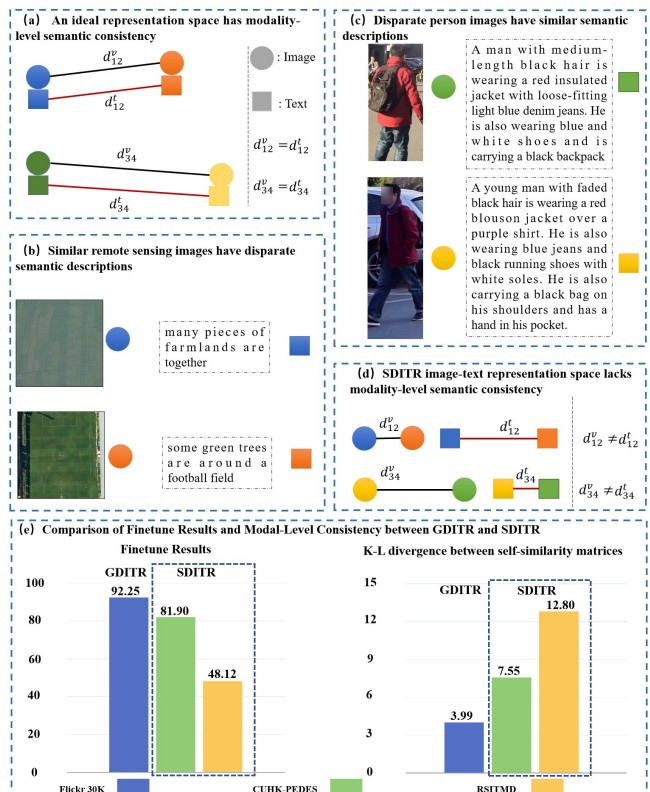

**Figure 1: (a) Ideal retrieval space with aligned intra-modal distance distributions. (b) High variability in text descriptions for similar remote sensing images. (c) Non-distinctive text descriptions for different pedestrian images. (d) RSITMD and TIReID exhibit poor modal-level distribution consistency. (e) Finetuning performance and modal-level distribution consistency are compared between GDITR and SDITR. Lower KL divergence indicates stronger consistency, vice versa.**

conduct an in-depth examination of SPDS to delineate its performance thresholds and the trade-off dynamics between self-pruning precision and model capacity. Our comparative analysis of SPDS with conventional teacher-student distillation models offers critical insights and serves as a reference for future research endeavors aimed at refining the efficiency of CLIP.

Our contributions are concisely summarized as follows:

- The Accurate and Lightweight learning framework for domain-specific image-text Retrieval (AIR) is introduced, wherein CLIP's robust features are leveraged to balance accuracy and computational cost efficiently in domain-specific image-text retrieval (SDITR) tasks.
- The Modal-Level distribution Consistency Enhancement regularization (MLCE) loss is introduced to align image-text modalities within CLIP, essential for optimizing joint representations in SDITR.
- The Self-Pruning Distillation Strategy (SPDS) has been devised, which reduces the computational requirements and

model size of CLIP without significantly compromising retrieval capabilities.
- Extensive experiments on diverse datasets validate MLCE loss's effectiveness, achieving new performance benchmarks. SPDS's trade-off between retrieval accuracy and model size has been explored. Comparative studies with traditional distillation methods provide insights for advancing efficient CLIP model reduction.

## 2 RELATED WORK

### 2.1 Specific Domain Image-Text Retrieval (SDITR)

SDITR refers to image-text retrieval tasks within certain industries and specialized fields. This paper focuses on two SDITR tasks: Remote Sensing Image-Text Retrieval (RSITR) and Text-Image person Re-identification (TIReID).

**RSITR** is a bi-directional retrieval technique that has become increasingly prominent due to its efficacy in managing and analyzing the growing volume of remote sensing (RS) imagery [8, 36, 49, 70, 72, 74, 75]. It leverages semantic similarities to retrieve RS images or corresponding text descriptions. Significant advancements in RSITR include a semantic alignment module by Cheng et al. [8] that employs attention and gating mechanisms for matching RS images with text. Yuan et al. have contributed extensively, introducing a fine-grained RS image-text dataset and frameworks that enhance retrieval accuracy [72] and support cross-modal retrieval encompassing images, text, and audio [74]. Other notable works include graph neural network models for feature interaction learning [70], unsupervised contrastive hashing for cross-modal retrieval [39], and multilingual frameworks [49] for RSITR.

Our study investigates the application of CLIP for RSITR, diverging from approaches like Yuan et al.'s parameter-efficient training [71]. Instead, we focus on preserving CLIP's performance while reducing parameter count and computational expense during inference.

**TIReID** aims to retrieve the most relevant individual from a large image repository using text query, with applications ranging from personal photo album searches to public safety. The task was initially proposed by Li et al. [29] along with the first benchmark dataset, CUHK-PEDES [29]. Subsequent methods [28] utilized VGG and LSTM as feature extractors for images and text, respectively, aligning them with a matching loss function. Later, studies [50, 80] improved feature extraction backbones with ResNet50/101 and BERT, and designed novel cross-modal matching losses to align global image-text features in a joint embedding space. Recent work has also incorporated CLIP. Yan et al. [66] introduced a CLIP-driven fine-grained information mining framework to transfer CLIP's knowledge. Jiang et al. [19] designed an implicit relational reasoning and alignment method to fine-tune CLIP.

Prior methods, however, did not consider reducing CLIP's data flow. Our paper conducts an in-depth investigation into this aspect, addressing the need for efficiency in TIReID systems.

### 2.2 Visual-Language large-scale Pre-training model (VLP)

The surge in VLP model development [17, 24, 26, 27, 30, 33, 34, 46, 77] has been fueled by breakthroughs in both language [9] and vision models [14], bolstered by expansive image-text datasets. Pre-training on auxiliary tasks like Image-Conditioned Masked Language Modeling and Image-Text Matching [46] equips these models for diverse applications, including retrieval and visual question answering. Key models include Lu et al.'s VILBERT [33], which leverages dual-stream architecture, and LXMERT by Tan et al. [53], which integrates object location data. Su et al. Li et al. [26] present stacked Transformer models, unifying object, image, and text features. Li et al. [30] and Kim et al. [24] further refine these models with specialized input structures and task optimizations. Zeng et al.'s X-VLM [77] and Jia et al.'s billion-pair model [17] continue this trend, emphasizing feature alignment and large-scale pre-training. Among these, CLIP [46] stands out for its simplicity and effectiveness, pre-trained from scratch on a vast internet-sourced dataset. It has become a benchmark for image-text tasks due to its straightforward architecture and contrastive learning approach. Our work builds upon CLIP's framework to enhance SDITR, focusing on achieving precision and speed in this specialized domain.

### 2.3 Knowledge Distillation

Knowledge distillation streamlines large-scale models (teacher models) into smaller, efficient counterparts (student models), a concept introduced by Bucilua et al. [3] and advanced by Hinton et al. [16]. It has since become a cornerstone for model compression, with extensive research contributions in unimodal [1, 15, 57, 63] and multimodal distillation [38, 43, 55, 62, 81].

**Unimodal Model Distillation:** Wu et al. [63] implemented distillation in open set domains, introducing angular and embedding constraints for recognition tasks. He et al. [15] tailored distillation for semantic segmentation, optimizing latent feature similarity with an autoencoder to align student-teacher features. Wang et al. [57] enhanced student networks in facial recognition through positional-aware exclusivity. Aguilar et al. [1] showcased distillation's efficacy using BERT's internal representation for language benchmarks.

**Multimodal Model Distillation:** Wang et al. [55] harnessed multimodal data via independent unimodal teacher models to instruct a multimodal student model. Zhao et al. [81] innovated a cross-modal scheme, allowing student training on datasets inaccessible to teachers. Miech et al. [38] mediated the speed-accuracy dichotomy in image-text retrieval with a Transformer-based teacher model. Several studies have optimized the CLIP model through distillation. TinyCLIP by Wu et al. [62] applied affinity imitation and weight inheritance for a more compact CLIP model. Pei et al. [43] introduced CLIPPING, a layer-wise technique using CLIP's visual encoder to refine MobileViT-v2.

Our work diverges by featuring self-distillation with SPDS, leveraging in-network deep-layer insights to inform shallow layers, thus bypassing the need for external student models and intricate distillation protocols, offering a more innovative approach.

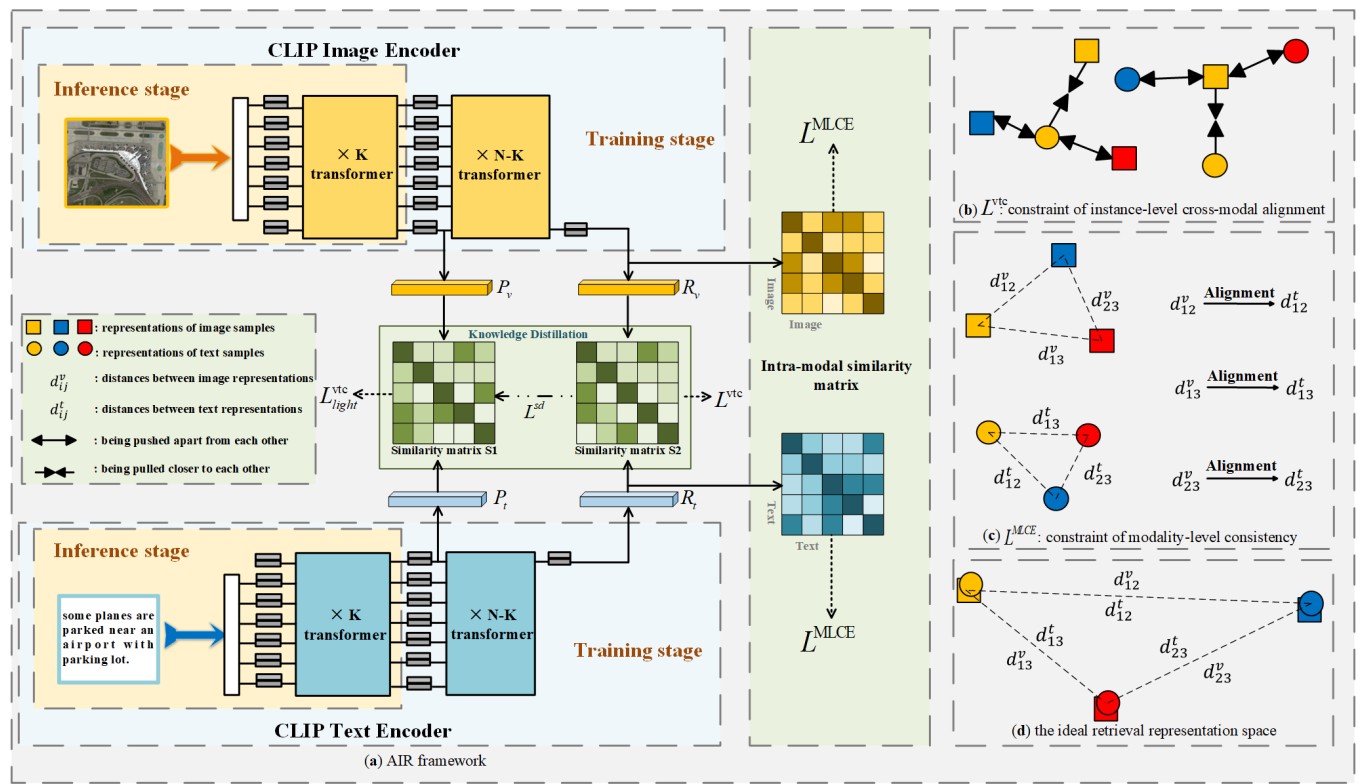

Figure 2: The pipeline of the proposed AIR framework. (a) introduces AIR framework, (b) and (c) respectively illustrate the effects of contrastive loss and MLCE loss, and (d) depicts the ideal feature distribution in the retrieval space.

## 3 METHOD

In this study, we present our framework from three aspects. Firstly, we introduce the MLCE loss. This innovation aims to enhance the consistency of modality sample distribution at the modality level. Then, we present the SPDS, which further enhances the test efficiency of our framework. Furthermore, we provide a comprehensive review of CLIP in Supplementary Materials.

### 3.1 Modal-Level Consistency distribution Enhancement (MLCE) loss

In this paper, we propose an MLCE loss based on CLIP to ensure the convergence of similarity distributions between samples in the text modality and the image modality. The MLCE loss imposes regularization constraints on the internal representations of the image and text modalities, which goes beyond the capability of the inherent contrastive loss in CLIP. The contrastive loss only enforces cross-modal consistency between images and text, as shown in Figure 2 (b). By incorporating the MLCE loss, we address the limitations of the contrastive loss and tackle the issue of poor modal-level distribution consistency in SDITR. This leads to a joint image-text representation space that approaches the ideal retrieval space, as illustrated in Figure 1 (a) and Figure 2 (d).

Specifically, we first compute the self-similarity matrices within the text and image modalities respectively, denoted as $\mathbf{S_t}$ and $\mathbf{S_v}$, as shown in Equations 1 and 2.

$$\mathbf{S_t} = 0.5 * (1 + \mathbf{R_t R_t}^T) \tag{1}$$

$$\mathbf{S_v} = 0.5 * (1 + \mathbf{R_v R_v}^T) \tag{2}$$

$\mathbf{R_v} = \{R_v^i, i = 1, 2, \ldots\}$ is a matrix of image feature vectors output by CLIP within a batch. $\mathbf{R_t} = \{R_t^i, i = 1, 2, \ldots\}$ denotes a matrix of text feature vectors within a batch. $T$ denotes the transposition operation.

Then, we calculate the K-L divergence between the text self-similarity matrix and the image self-similarity matrix to use as the MLCE loss. The MLCE loss is shown in Equation 3,

$$L^{MLCE} = \frac{1}{m} \sum_{i=1}^{m} KL(Softmax(\frac{\mathbf{S_t}^i}{\mu}) \| Softmax(\frac{\mathbf{S_v}^i}{\mu})) \tag{3}$$

where $L^{MLCE}$ denotes the MLCE loss, $KL()$ denotes the K-L divergence, $m$ denotes the batch size, and $\mu$ denotes the temperature coefficient.

The total loss of the model is shown in Equation 4,

$$L^{total} = L^{vtc} + \alpha L^{MLCE} \tag{4}$$

where $L^{vtc}$ denotes the cross-modal contrastive loss used in CLIP, $L^{total}$ is the total loss, and $\alpha$ represents the combination coefficient.

## 3.2 Self-Pruning Distillation Strategy (SPDS)

Although CLIP with $L^{MLCE}$ can achieve SDITR, it has a large number of parameters and incurs high computational costs during inference. Consequently, processing massive RS images in practical scenarios becomes unfeasible. To alleviate this issue, a novel SPDS is designed to lighten CLIP.

As shown in Figure 2, we use the output of the $K$-th Transformer block of CLIP as the lightweight image and text features within a batch, denoted as $\mathbf{P_v}$ and $\mathbf{P_t}$, respectively. We then calculate the similarity matrix $\mathbf{S_1}$ between the $P_v$ and $P_t$. The similarity matrix of the raw outputs from the last Transformer block of CLIP is denoted as $\mathbf{S_2}$.

$$\mathbf{S_1} = \mathbf{P_v}\mathbf{P_t}^T \tag{5}$$

$$\mathbf{S_2} = \mathbf{R_v}\mathbf{R_t}^T \tag{6}$$

$T$ denotes the transposition operation.

During the optimization, we use $L_{light}^{vtc}$ and $L^{vtc}$ for $\mathbf{S_1}$ and $\mathbf{S_2}$ respectively. Both of $L_{light}^{vtc}$ and $L^{vtc}$ represent the cross-modal contrastive losses. Additionally, a novel self-distillation loss $L^{sd}$ is proposed, so that the information of $\mathbf{S_2}$ can guide $\mathbf{S_1}$ in optimization, thus achieving knowledge transfering.

$$
\begin{aligned}
L^{sd} = (& -\sum_{i=1}^{m}\sum_{j=1}^{m} \frac{exp\left(\mathbf{S_2}^{ij}/\gamma\right)}{\sum_{k=1}^{m} exp\left(\mathbf{S_2}^{ik}/\gamma\right)} \\
& log(\frac{exp\left(\mathbf{S_1}^{ij}/\gamma\right)}{\sum_{k=1}^{m} exp\left(\mathbf{S_1}^{ik}/\gamma\right)})) + \\
(& -\sum_{i=1}^{m}\sum_{j=1}^{m} \frac{exp\left(\mathbf{S_2^T}^{ij}/\gamma\right)}{\sum_{k=1}^{m} exp\left(\mathbf{S_2^T}^{ik}/\gamma\right)} \\
& log(\frac{exp\left(\mathbf{S_1^T}^{ij}/\gamma\right)}{\sum_{k=1}^{m} exp\left(\mathbf{S_1^T}^{ik}/\gamma\right)}))
\end{aligned}
\tag{7}
$$

where $m$ denotes the batch size and $\gamma$ denotes the temperature coefficients.

With the inclusion of lightweight, we modify the overall loss in Equation 4 to the following form:

$$L^{total} = L^{vtc} + L_{light}^{vtc} + \alpha L^{MLCE} + \eta L^{sd} \tag{8}$$

where $\alpha$ and $\eta$ represents the combination coefficients.

During inference, we only keep the former K Transformer blocks, which greatly reduces the parameters capacity and computation of the test.

## 4 EXPERIMENTS

To demonstrate the efficacy of the AIR framework, we executed comprehensive experiments in RSITR and TIReID. We describe the experimentation protocols and datasets initially, followed by comparative analyses highlighting AIR's retrieval efficacy, parameter efficiency, and computational economy. Ablation studies further elucidate the MLCE loss and the layer-wise influence of the SPDS.

Additional experiments contrast SPDS with conventional teacher-student knowledge distillation approaches.

## 4.1 Implementation details, Metrics, and datasets

Our experiments were performed on an NVIDIA RTX 3090 GPU within the PyTorch framework using the AdamW optimizer. We trained our model for 10 epochs on the RSITR task and 30 epochs on the TIReID task, employing a batch size of 64 and an initial learning rate of 1e-5, with learning rate decay managed by a cosine annealing strategy. We utilize the pretrained model weights from CLIP as the initialization for our model. We strictly followed the official train-test splits for all datasets. We conducted an exhaustive hyperparameter search for each parameter, such as the combination coefficients $\alpha$, $\eta$, and the temperature coefficient $\mu$ in the loss functions. Details of these experiments can be found in the Supplementary Material. The hyperparameters in the main experiments were all set based on the outcomes presented in the Supplementary Material.

Following established benchmarks [72], for RSITR, we measure performance using recall rates R@k (k=1,5,10) and mean Recall (mR) for both text-to-image and image-to-text tasks. R@k quantifies the percentage of correct results in the top k retrievals, while mR is the average of the six R@k values. For TIReID [19], we utilize R@K for text-to-pedestrian tasks and mean Average Precision (mAP) for overall performance evaluation. Additionally, we assess the model's efficiency by computing the Parameters and Floating Point Operations (FLOPs) **at inference**.

We employed the following four **RSITR** datasets. RSITMD [72] features 4,743 images from 32 land-cover classes, each paired with text. RSICD [35] comprises 10,921 images across 31 categories. UCM Caption [44] contains 2,100 images over 21 scenes. Sydney Caption [44] includes 613 images in 7 categories. Each image in these datasets is coupled with five textual descriptions.

We investigated the following two **TIReID** datasets. CUHK-PEDES [29] encompasses 40,206 images of 13,003 subjects, with 80,412 text descriptions. RSTPReid [85] contains 20,505 images of 4,101 individuals from 15 camera views, each with two text descriptions.

## 4.2 Comparison experiments

*4.2.1 RSITR Results.* To highlight our method's superiority, we conducted comparative experiments against a spectrum of established techniques. Results were compiled from existing literature and complemented by our reimplementations. These techniques are classified into three main groups. The initial group encompasses conventional image-text retrieval and RSITR methodologies. The second group is formed by several Transformer-based methods. The third group is composed of CLIP-based variants. We pitted various configurations of our proposed method against these comparative methods. "AIR(w/o SPDS)" denotes the performance of the fine-tuned CLIP model with our MLCE loss on the RSITR dataset, absent the SPDS. "AIR(k=3)" and "AIR(k=9)" reflect the outcomes with the application of SPDS, resulting in pruned models with 3 and 9 layers, respectively. We conducted comparative experiments on the RSICD, RSITMD, and UCM Caption datasets, with the RSICD

## Table 1: Experimental results on RSITR

| Dataset | | | RSITMD | | | | | | | RSICD | | | | | | |
| Method | Test Parameters (M) | Test FLOPS (G) | Sentence Retrieval | | | Image Retrieval | | | mR | Sentence Retrieval | | | Image Retrieval | | | mR |
| | | | R@1 | R@5 | R@10 | R@1 | R@5 | R@10 | | R@1 | R@5 | R@10 | R@1 | R@5 | R@10 | |
| *Traditional methods* | | | | | | | | | | | | | | | | |
| SCAN [25] ECCV'18 | 13.68 | 2.42 | 11.06 | 25.88 | 39.38 | 9.82 | 29.38 | 42.12 | 26.28 | 5.85 | 12.89 | 19.84 | 3.71 | 16.4 | 26.73 | 14.23 |
| CAMP [58] ICCV'19 | 36.64 | 2.28 | 11.73 | 26.99 | 38.05 | 8.27 | 27.79 | 44.34 | 26.20 | 5.12 | 12.89 | 21.12 | 4.15 | 15.23 | 27.81 | 14.39 |
| CAMERA [45] MM'20 | 131.81 | 4.21 | 8.33 | 21.83 | 33.11 | 7.52 | 26.19 | 40.72 | 22.95 | 4.57 | 13.08 | 21.77 | 4.00 | 15.93 | 26.97 | 14.39 |
| AMFMN [72] TGRS'22 | 35.94 | 2.75 | 11.06 | 29.20 | 38.72 | 9.96 | 34.03 | 52.96 | 29.32 | 5.39 | 15.08 | 23.40 | 4.90 | 18.28 | 31.44 | 16.42 |
| MCRN [74] JAG'22 | 52.35 | 4.87 | 13.27 | 29.42 | 41.59 | 9.42 | 35.53 | 52.74 | 30.33 | 6.59 | 19.40 | 30.28 | 5.03 | 19.38 | 32.99 | 18.95 |
| LW-MCR [73] TGRS'21 | 1.65 | 0.46 | 9.73 | 26.77 | 37.61 | 9.25 | 34.07 | 54.03 | 28.58 | 4.39 | 13.35 | 20.29 | 4.30 | 18.85 | 32.34 | 15.59 |
| GALR [76] TGRS'22 | 46.89 | 2.57 | 14.82 | 31.64 | 42.48 | 11.15 | 36.68 | 51.68 | 31.41 | 6.59 | 19.85 | 31.04 | 4.69 | 19.48 | 32.13 | 18.96 |
| SWAN [54] ICMR'23 | 37.50 | 6.54 | 13.35 | 32.15 | 46.90 | 11.24 | 40.40 | 60.60 | 34.11 | 7.41 | 20.13 | 30.86 | 5.56 | 22.26 | 37.41 | 20.61 |
| HVSA [79] TGRS'23 | 35.01 | 2.51 | 13.20 | 32.08 | 45.58 | 11.43 | 39.20 | 57.45 | 33.16 | 7.47 | 20.62 | 32.11 | 5.51 | 21.13 | 34.13 | 20.16 |
| PIR [42] MM'23 | – | – | 18.14 | 41.15 | 52.88 | 12.17 | 41.68 | 63.41 | 38.24 | 9.88 | 27.26 | 39.16 | 6.97 | 24.56 | 38.92 | 24.26 |
| MGRM [78] TGRS23' | – | – | 13.51 | 31.87 | 46.27 | 11.11 | 37.22 | 56.61 | 32.76 | 7.41 | 23.24 | 35.32 | 5.75 | 21.23 | 35.55 | 21.42 |
| SMLGN [7] TGRS'24 | – | – | 17.26 | 39.38 | 51.55 | 13.19 | 43.94 | 60.40 | 37.62 | 8.87 | 25.53 | 37.24 | 7.85 | 27.14 | 42.58 | 24.87 |
| *Additional variants* | | | | | | | | | | | | | | | | |
| VIT+BERT | 171.29 | 19.50 | 12.83 | 31.19 | 46.24 | 9.60 | 36.59 | 54.42 | 31.81 | 9.06 | 22.78 | 32.75 | 5.32 | 19.47 | 33.71 | 20.52 |
| ResNet18 + BERT | 97.28 | 4.46 | 16.37 | 31.19 | 42.04 | 9.73 | 33.76 | 51.59 | 30.78 | 7.23 | 22.05 | 34.58 | 4.54 | 19.25 | 33.5 | 20.19 |
| ResNet101+ BERT | 129.13 | 10.50 | 13.50 | 32.30 | 46.24 | 11.90 | 36.46 | 52.43 | 32.14 | 9.15 | 23.7 | 35.32 | 5.07 | 19.69 | 33.21 | 21.02 |
| *CLIP based methods* | | | | | | | | | | | | | | | | |
| CLIP-zero-shot ICML'21 | 82.46 | 13.21 | 9.51 | 25.00 | 33.41 | 7.79 | 28.98 | 45.66 | 25.06 | 7.23 | 17.38 | 26.26 | 5.38 | 18.12 | 29.17 | 17.26 |
| CLIP-full-finetune ICML'21 | 82.46 | 13.21 | 25.88 | 50.22 | 63.27 | 23.14 | 56.11 | 72.74 | 48.56 | 19.21 | 38.15 | 50.59 | 14.07 | 38.50 | 54.40 | 35.82 |
| Maple [23] CVPR'23 | 87.24 | 13.21 | 21.46 | 40.26 | 54.86 | 15.53 | 47.35 | 67.92 | 41.23 | 13.08 | 30.74 | 41.08 | 10.65 | 32.08 | 48.22 | 29.31 |
| CoOp [84] IJCV'22 | 82.47 | 13.21 | 9.73 | 25.22 | 39.82 | 8.19 | 32.39 | 51.37 | 27.85 | 7.87 | 21.32 | 31.47 | 6.24 | 21.70 | 34.44 | 20.51 |
| VPT [18] ECCV'22 | 82.55 | 13.21 | 13.72 | 34.29 | 48.67 | 13.72 | 40.89 | 59.56 | 34.14 | 9.97 | 24.52 | 37.33 | 9.64 | 29.49 | 44.76 | 25.95 |
| *Ours* | | | | | | | | | | | | | | | | |
| AIR (w/o SPDS) | 2.46 | 13.21 | **29.20** | **49.78** | **65.27** | **26.06** | **57.04** | **73.98** | **50.22** | **18.85** | **39.07** | **51.78** | **14.24** | **39.03** | **54.49** | **36.24** |
| AIR (k=9) | 61.99 | 9.94 | 23.67 | 44.47 | 57.3 | 19.96 | 51.73 | 69.82 | 44.49 | 14.55 | 33.58 | 45.93 | 11.03 | 32.94 | 49.62 | 31.28 |
| AIR (k=3) | 21.86 | 3.39 | 14.82 | 33.85 | 46.2 | 11.19 | 37.88 | 57.92 | 33.65 | 7.69 | 20.77 | 33.58 | 5.27 | 20.68 | 36.29 | 20.71 |

## Table 2: Experimental results on TIReID

| method | R@1 | R@5 | R@10 | mAP | Test FLOPs (G) | Test Parameters (M) | Ref |
| --- | --- | --- | --- | --- | --- | --- | --- |
| *Traditional methods* | | | | | | | |
| MANET [67] | 63.92 | 82.15 | 87.69 | – | 11.14 | 83.75 | TNNLS 23 |
| TIMAM [50] | 54.51 | 77.56 | 84.78 | 35.13 | 11.68 | 63.96 | ICCV 19 |
| HGAN [82] | 59.00 | 79.49 | 86.62 | 37.80 | 5.26 | 120.84 | MM 20 |
| C2A2 [40] | 64.82 | 83.54 | 89.77 | – | 12.87 | 107.71 | MM 22 |
| RKT [64] | 61.48 | 80.74 | 87.28 | – | 7.16 | 63.34 | TMM 23 |
| PWM-ATH [6] | 27.14 | 49.45 | 61.02 | – | – | 137.61 | WACV 18 |
| Dual-Path [83] | 44.40 | 66.26 | 75.07 | – | – | 185.73 | TOMM 20 |
| MIA [41] | 53.10 | 75.00 | 82.90 | – | – | 211.00 | TIP 20 |
| PMA [20] | 53.81 | 73.54 | 81.23 | – | 55.29 | 157.46 | AAAI 20 |
| DCMG [37] | 55.81 | 77.44 | 84.87 | – | 14.82 | 134.86 | IVC 21 |
| LCR^2S [65] | 67.36 | 84.19 | 89.62 | 59.24 | 4.64 | 87.82 | MM 23 |
| ILTL [4] | 67.13 | 84.60 | 90.37 | – | 24.02 | 82.46 | CVPR 23 |
| Unipt [51] | 66.83 | 84.16 | 89.42 | – | 13.59 | 142.91 | ICCV 23 |
| IVT [52] | 65.59 | 83.11 | 89.21 | – | 19.07 | 142.32 | ECCVW 22 |
| TGDA [12] | 64.64 | 83.38 | 89.34 | 58.64 | 5.08 | 113.84 | TCSVT 23 |
| IMG-NET [61] | 56.48 | 76.89 | 85.01 | – | – | – | JEI 20 |
| DSSL [85] | 59.98 | 80.41 | 87.56 | – | 4.70 | 53.89 | MM 21 |
| SUM [59] | 59.22 | 80.35 | 87.60 | – | 5.29 | 39.11 | KBS 22 |
| LBUL [60] | 64.04 | 82.66 | 87.22 | – | 8.88 | 57.99 | MM 22 |
| *CLIP based methods* | | | | | | | |
| CLIP-full-finetune | 67.37 | 86.52 | 91.88 | 60.73 | 13.21 | 82.46 | ICML 21 |
| CLIP-zero-shot | 12.64 | 27.13 | 5.56 | 11.15 | 13.21 | 82.46 | ICML 21 |
| *Ours* | | | | | | | |
| AIR (w/o SPDS) | **68.83** | **86.34** | **91.60** | **61.48** | 13.21 | 82.46 | – |
| AIR (k=9) | 62.52 | 81.17 | 87.51 | 57.93 | 9.94 | 61.99 | – |

and RSITMD results in Table 1 and UCM Caption findings in the Supplementary Material.

On the RSITMD dataset, our AIR (w/o SPDS) achieved a SOTA mR of 50.22. The AIR (k=9) and AIR (k=3) configurations attained mR scores of 44.49 and 33.65, respectively, while significantly reducing computational and parameter costs. Our method outperforms traditional methods, with our AIR (w/o SPDS or k=9) substantially surpassing the previous best mR of 38.24 by PIR. Compared to additional variants, our configurations exceed the mR of ResNet (VIT)

+ BERT, with the added benefit of a smaller parameter footprint and lower computational costs. Our method's superiority is evident when compared to CLIP-based methods. AIR (w/o SPDS) surpasses CLIP (full-finetune) by 1.66 in mR without increasing test-time resources. AIR (k=9) offers a balance, reducing resources while maintaining a competitive mR of 44.49. Compared to the recent prompt-based Maple [23], our AIR (k=9) not only achieves higher mR scores but does so with reduced computational complexity and parameters.

On the RSICD dataset, our method also sets a new state-of-the-art, with AIR (w/o SPDS), AIR (k=9), and AIR (k=3) achieving mR values of 36.24, 31.28, and 20.71, respectively, outperforming all traditional methods. In terms of resource efficiency, our lighter models still deliver competitive results. Even our lightweight setting AIR (k=9) surpasses the highest traditional method KAMCAL. Against additional variants, our configurations perform notably better, with AIR (k=9) improving upon ResNet101 + BERT and VIT + BERT by 10.26 and 10.76 in mR, respectively, while requiring less computation and fewer parameters. When benchmarked against other CLIP-based approaches, our models demonstrate enhanced performance. AIR (w/o SPDS) edges out CLIP (full-finetune) by 0.42 in mR, while AIR (k=9) achieves a good balance between performance and efficiency, underscoring the effectiveness of our SPDS in achieving a lightweight yet high-performing network.

*4.2.2 TIReID Results.* We performed comparative experiments for TIReID on the CUHK PEDES and RSTPReid datasets. The results on CUHK PEDES are shown in Table 2, and the results for RSTPReid can be found in the Supplementary Materials. We categorized the comparison algorithms into traditional methods and CLIP-based methods. As shown in Table 2, AIR without SPDS achieved an R@1

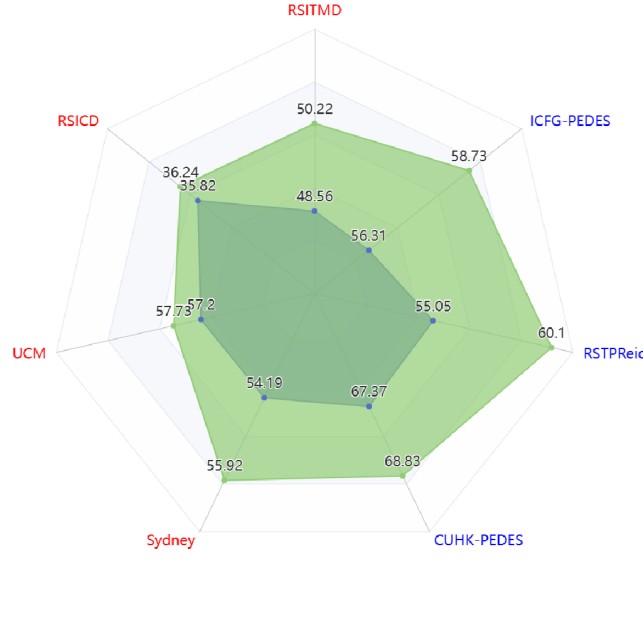

**Figure 3: Efficetness of MLCE loss. Red font denotes RSITR datasets, while blue indicates TIReID datasets.**

score of 68.83, surpassing all traditional methods. Compared to CLIP-based methods, AIR without SPDS exceeded the performance of CLIP-full-finetune by 1.46 in R@1, demonstrating the effectiveness of MLCE loss. Our lightweight version of AIR (k=9) achieved an R@1 of 62.52, maintaining competitive retrieval precision while reducing computational and parameter requirements, which evidences that SPDS can achieve a favorable balance between retrieval accuracy and speed.

### 4.3 Abalation study

*4.3.1 Efficetness of MLCE loss.* We have demonstrated the effectiveness of the MLCE loss on a variety of RSITR and TIReID datasets. A visual representation of our findings is depicted in Figure 3, which clearly illustrates the superior performance of our proposed loss function. For each dataset involved in these two distinct SDITR tasks, the use of the MLCE loss yields better results than merely fine-tuning the CLIP model.

When employing the MLCE loss in conjunction with the CLIP model on the RSITMD dataset, we observe a mR improvement of 1.66 points over the baseline of a fine-tuned CLIP, all the while maintaining an equal computational cost and model size during the evaluation phase. This underscores the efficacy of our MLCE loss. On the RSICD, integrating MLCE loss with CLIP results in an mR increase of 0.42, further evidencing the utility of our loss function. Moreover, the application of MLCE loss with CLIP on the UCM Caption and Sydney datasets leads to mR enhancements of 0.53 and 1.73, respectively. The improvements are even more pronounced in the TIReID tasks. When applying MLCE loss, we achieve substantial gains in R@1, with increases of 1.46 on the

CUHK-PEDES dataset, 5.05 on the RSTPReid dataset, and 2.42 on the ICFG-PEDES dataset.

To further investigate MLCE loss's impact, we conducted supplementary parameter search for $\mu$ and $\alpha$ in Equations 3 - 4, identifying optimal settings as detailed in the Supplementary Material.

*4.3.2 Ablation of SPDS.* We performed a search on the combination coefficient $\eta$ of self-distillation loss, with $K = 3$ and $\gamma = 8$ fixed empirically. The experimental results are shown in Figure 4. We set the combination coefficient $\eta$ to the following values: {0.5, 0.1, 0.05, 0.01, 0.005, 0.001, 0.0005, 0.0001, 0.00005, 0.00001}. From Figure 4, it is evident that when $\eta$ is too small, the mR value decreases, indicating that our designed self-distillation loss is beneficial for learning shallow CLIP features and facilitates the transfer of knowledge from deeper to shallower layers. Conversely, when $\eta$ is too large, the mR value also decreases, due to the excessive $\eta$ impeding the optimization of the contrastive loss. The optimal value of mR is achieved when $\eta$ equals 0.1.

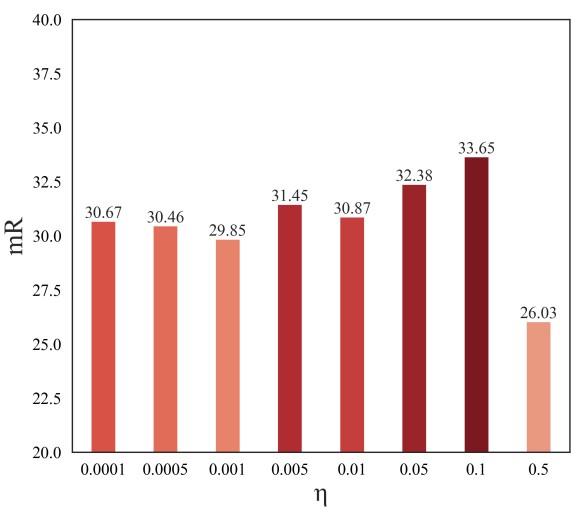

**Figure 4: Search on the combination coefficient $\eta$ of self-distillation loss**

To assess the impact of layer count on the effectiveness of SPDS, a comprehensive optimization was conducted to determine the optimal values for the number of SPDS layers, $K$, and the temperature coefficient, $\gamma$. This optimization was performed over six datasets for RSITR and TIReID. Here, we present the results on the RSICD dataset within RSITR. Results for other datasets can be found in the Supplementary Material. For the RSITR task, the mR was employed as the metric for joint search. We set $K$ to the set {3, 4, 5, 6, 7, 8, 9, 10} and $\gamma$ to {2, 4, 6, 8, 10}. The resultant performance on the RSICD dataset is illustrated in Figure 5. As depicted in Figure 5, a $K$ value of 3 enables our self-distillation approach to achieve an mR of 20.71 on the RSICD dataset, outperforming the majority of traditional methods. Moreover, a general trend of increasing mR with the rising number of layers was observed, aligning with expectations.

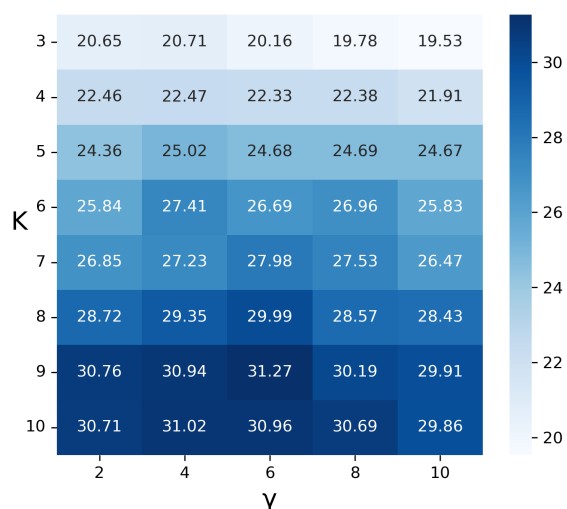

**Figure 5: Joint search results for $K$ and $\gamma$ on RSICD**

The investigation highlighted the significant efficiency improvements realized by the SPDS, which markedly reduces the parameter and computational overheads associated with large-scale models, without incurring substantial losses in retrieval accuracy. An analysis of SPDS (elucidated in the Supplementary Materials) delineates a direct correlation between the number of blocks, $K$, and both parameter count and computational complexity. The integration of SPDS results in pronounced reductions in these metrics, especially when juxtaposed with CLIP-based methods, while still maintaining competitive retrieval performance.

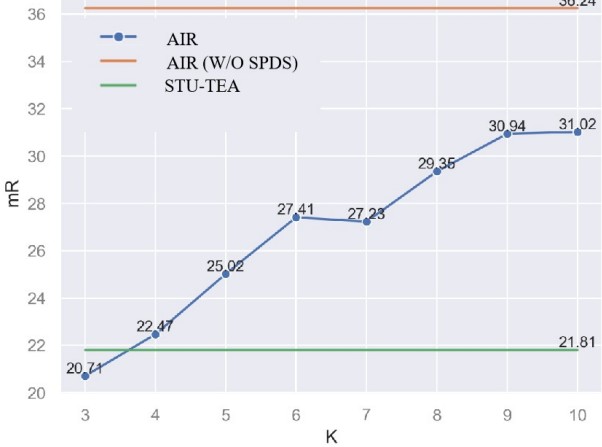

**Figure 6: Curve of mR variation with K on RSICD**

### 4.4 Futher Exploration

To further explore the lightening of CLIP in RSITR, we also conducted experiments using the student-teacher distillation method based on [31]. We design a lightweight student model based on MobileNet and TextCNN architectures. The teacher model is our "AIR(w/o SPDS)". For the design and specific details of the student model, please refer to the Supplementary Material.

The "STU-TEA" represents the results of the student-teacher distillation approach. We can observe that STU-TEA achieves mR values of 21.81 on the RSICD. The performance of STU-TEA surpasses the traditional comparison methods. Furthermore, the student model in STU-TEA has significantly lower parameters capacity and computational complexity, with only 0.09G and 9.15M, respectively, which is much smaller compared to other methods. We examine the variation of mR values concerning K for SPDS on the RSITMD and RSICD, comparing AIR with STU-TEA. The experimental results on the RSITMD are shown in Figure 6. For the experimental results on the RSITMD, please refer to the Supplementary Material. We observed that as K increases, the mR values for AIR also increase.when K is greater than 3, AIR achieves higher mR values than STU-TEA. Additionally, AIR(W/O SPDS) with 12 Transformer blocks has the highest mR value, reaching 36.24. These observations demonstrate the strong flexibility of our proposed method. When users prioritize high mR values without considering computational speed and memory consumption, AIR(W/O SPDS) can be chosen. When users face strict requirements for computational speed and memory limitations, STU-TEA can be selected. Finally, when users consider a trade-off between computational speed and retrieval accuracy (mR), the value of K can be determined based on their specific needs to utilize AIR.

## 5 CONCLUSION

In this paper, we address the adaptation of pre-trained cross-modal Visual-Language models for Specific Domain Image-Text Retrieval (SDITR), tackling modal-level distribution inconsistency and excessive computational demands at inference. We introduce a novel Modal-Level distribution Consistency Enhancement regularization (MLCE) loss, derived from the CLIP framework, to harmonize image-text representations. Furthermore, we propose a Self-Pruning Distillation Strategy (SPDS) to mitigate parameter and computational overhead during testing. This strategy utilizes the CLIP model's cross-modal output to refine shallower-layer learning, ensuring only the essential layers are retained for a compact inference model. Empirical evaluations across diverse datasets confirm that our MLCE loss significantly advances joint representation, leading to unparalleled retrieval accuracy in remote sensing image-text retrieval and text-image person re-identification tasks. SPDS is extensively tested, demonstrating its effectiveness in balancing accuracy with computational efficiency. We also examine the limits of SPDS's impact and its trade-off between accuracy and model size, and compare SPDS to conventional teacher-student knowledge distillation methods.

Despite these contributions, our study is limited to the CLIP model. Further research is needed to assess the applicability of our strategies across a wider spectrum of VLPs. Future work will focus on expanding the generalizability of our findings.

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
