# OpenReview forum: "Accurate and Lightweight Learning for Specific Domain Image-Text Retrieval"
_acmmm.org/ACMMM/2024/Conference — MM2024 Poster_

### Official Review · Reviewer_tr8t · 2024-05-17

**Rating:** 3
**Confidence:** 4

**Summary:**

This paper addresses two issues in image-text retrieval tasks applied to specific domains: suboptimal retrieval spaces and unnecessarily high inference computational loads. The authors propose a novel framework, Accurate and Lightweight learning for specific domain Image-text Retrieval (AIR), based on the CLIP architecture. AIR incorporates a Modal-Level Distribution Consistency Enhancement (MLCE) regularization loss and a Self-Pruning Distillation Strategy (SPDS). The MLCE loss harmonizes the sample distance distributions within image and text modalities. To achieve model lightweighting, SPDS employs a strategic knowledge distillation process to mitigate parameter and computational overhead during testing.

**Strengths:**

1. For balancing accuracy and computational cost efficiently in domain-specific image-text retrieval (SDITR) tasks, the authors introduce the Accurate and Lightweight learning framework for domain specific image-text Retrieval (AIR).
2. The Modal-Level distribution Consistency Enhancement (MLCE) loss is introduced to align image-text modalities within CLIP. Experiments conducted on the RSITR and TIReID datasets validate the effectiveness of the MLCE loss.

**Limitations:**

1. The experimental details are not sufficiently clear. The authors should specify the backbones used by different methods, and comparisons should be made under the same conditions, rather than simply comparing the final performance.
2. Ablation studies lack sensitivity analysis on factors such as batch size.
3. The paper lacks a comprehensive analysis of how the algorithm scales with increasing model size. Moreover, it does not evaluate other types of backbones besides CLIP-ViT-B/32, such as BLIP and BLIP-2. Conducting experiments with a variety of backbone architectures would provide a more thorough understanding of the algorithm's scalability and its performance across different model sizes.
4. In Eq. (8), the authors employ four losses to optimize the network, with different weights assigned to each loss. However, there are redundancies between the different types of losses, and having too many loss categories can make it challenging to control the loss weights effectively. This complexity might lead to difficulties in achieving optimal performance due to the intricate balancing required among the various losses.4. Inference efficiency is as important as Test FLOPS. The proposed AIR involves the calculation of two similarities (S1, S2). It is crucial to understand how these calculations impact the inference efficiency of the model. Therefore, the authors need to provide a detailed comparison of the inference efficiency.
5. What are the advantages of the proposed AIR compared with the currently popular pre-training models? Such as RemoteCLIP and GeoRSCLIP.
6. Section 2.2 should introduce the updated VLP model, which is currently somewhat outdated.
7. There is a writing inconsistency between the main text and the supplementary materials. The model is referred to as AIR in the main text, whereas it is referred to as ALR in the supplementary materials. This discrepancy can cause confusion and should be corrected for consistency.
8. The resolution of the images in the manuscript is insufficient, which can hinder the clarity and readability of important details.  To improve the visual quality and ensure that the images are clear and informative at any scale, it is recommended to use vector images.

**Suitability:**

3

---

### Official Review · Reviewer_MmLj · 2024-05-24

**Rating:** 4
**Confidence:** 2

**Summary:**

This paper proposes a new framework for Specific Domain Image-Text Retrieval (AIR) based on the CLIP architecture, focusing on accuracy and lightweight learning. AIR combines a Modal-Level Consistency Enhancement (MLCE) loss and a Self-Pruning Distillation Strategy (SPDS) to improve retrieval precision and computational efficiency. The MLCE loss aligns sample distance distributions within image and text modalities to create a more optimal retrieval space. Meanwhile, SPDS distills deep multimodal insights from CLIP into a shallower model, retaining only essential layers for inference, thereby lightening the model. The study also explores the limitations of SPDS and compares it with traditional teacher-student distillation methods.

**Strengths:**

1. MLCE loss and SPDS address key issues of alignment and real-time inference in specific domain image-text retrieval.

2. The paper clearly explains how MLCE loss aligns image and text modalities and the impact of contrastive learning loss.

3. SPDS offers flexibility, allowing users to balance retrieval accuracy and computational efficiency based on their specific needs.

**Limitations:**

1. The Method section is relatively brief and simplistic, lacking in-depth explanations that would provide a more thorough understanding of the implementation

2. the term "Efficentness" in Figure 3 is not clearly defined or quantified, leading to potential confusion about its measurement and implications

3. There appears to be a potential error in Table 1 regarding the test parameters for the w/o SPDS condition, as they do not correspond with the Test FLOPS values

4. Presentation Issues: Figures are not vector graphics and formulas lack proper punctuation.

**Suitability:**

3

---

### Official Review · Reviewer_wusG · 2024-05-26

**Rating:** 3
**Confidence:** 4

**Summary:**

This work develop a method for Remote Sensing Image-Text Retrieval (RSITR) and Text-Image Person Re-identification (TIReID). It presents a framework based on the CLIP architecture. AIR incorporates a Modal-Level distribution Consistency Enhancement regularization (MLCE) loss and a Self-Pruning Distillation Strategy (SPDS). The MLCE loss harmonizes the sample distance distributions within image and text modalities. Meanwhile, SPDS employs a strategic knowledge distillation process to transfer deep multimodal insights from CLIP to a shallower level, maintaining only the essential layers for inference, thus achieving model light-weighting.

**Strengths:**

The proposed method explores the limits of SPDS’s performance and compares it with conventional teacher-student distillation methods.
The findings reveal that MLCE loss secures optimal retrieval.

**Limitations:**

1. The motivation is unclear. Please give more explanation on "the enhancement of modal-level distribution consistency within the retrieval space resulting in suboptimal retrieval spaces”, and I suggest authors to provide some experimental results to demonstrate it.
2.  The performance of the MLCE loss and Self-Pruning Distillation Strategy (SPDS) should be analyzed and explained theoretically.
3. The proposed method is lightweight learning, please porvide experimental results on run time analysis.
4. Writings and arrangements of the paper need improvements.

**Suitability:**

2

---

### Official Review · Reviewer_wTHe · 2024-05-27

**Rating:** 5
**Confidence:** 3

**Summary:**

This paper proposes an  accurate and lightweight learning for specific domain image-text retrieval (AIR), based on the CLIP architecture. CLIP’s  features are leveraged to balance accuracy and computational cost efficiently in the domain-specific image-text retrieval task.

**Strengths:**

A Modal-Level distribution Consistency Enhancement regularization (MLCE) loss and a Self-Pruning Distillation Strategy (SPDS) are incorporated to improve retrieval precision and computational efficiency. The MLCE loss harmonizes the sample distance distributions within image and text modalities. SPDS employs a strategic knowledge distillation process to transfer deep multimodal insights from CLIP to a shallower level, maintaining only the essential layers for inference for model light-weighting. Comprehensive experiments demonstrate the effectiveness of both MLCE loss and SPDS.

**Limitations:**

Some points to improve:
1. There is a formatting issue: the spacing between sections 2.2 and 2.3 is a bit too large.、
2. When defining tensors, indicating their dimensions would help readers understand better.
3. A Self-Pruning Distillation Strategy (SPDS) is proposed to improve  computational efficiency. It is better to give the computional complexity or reference time for conviencement.
4. I suggest the authors to label the best results for previous state-of-the-art methods, for readers convenience, and give the definition of bold fond.
5. The work is not reproducible. The codes are not available, and there are no instructions.

**Suitability:**

3

---

### Meta-Review · Area_Chair_c8uj · 2024-07-02

**Recommendation:** Accept (Poster)
**Confidence:** 3

**Metareview:**

The reviewers are not in consensus regarding this paper. However, given the authors' rebuttal and the possibility that some issues will be solved in the final version, I suggest accepting the paper.
Reviewer wusG mentioned 4 limitations in his initial review. To 3 of them, the authors answered, and the reviewer didn't change his rating and didn't contest the answer. The first limitation was referring to a phrase that is not in the paper and the authors discussed it in the rebuttal. Thus, even if wusG confirmed the Borderling Reject, I tend towards acceptance also considering that his suggestion to reject was, in fact, borderline.
The only other reviewer suggesting rejection is tr8t, whose review was very detailed. However, after the rebuttal, the major concern is redundancies between the different types of losses and limited performance, to which the authors answered that the "approach focuses on optimizing spatial representations and model lightweight".